# Combined Sulfur and Peroxide Vulcanization of Filled and Unfilled EPDM-Based Rubber Compounds

**DOI:** 10.3390/ma16165596

**Published:** 2023-08-12

**Authors:** Ján Kruželák, Mária Mikolajová, Andrea Kvasničáková, Michaela Džuganová, Ivan Chodák, Ján Hronkovič, Jozef Preťo, Ivan Hudec

**Affiliations:** 1Department of Plastics, Rubber and Fibres, Faculty of Chemical and Food Technology, Slovak University of Technology in Bratislava, Radlinského 9, 812 37 Bratislava, Slovakia; maria.mikolajova@stuba.sk (M.M.); andrea.kvasnicakova@stuba.sk (A.K.);; 2Polymer Institute, Slovak Academy of Sciences, Dúbravská cesta 9, 845 41 Bratislava, Slovakia; 3VIPO a.s., Gen. Svobodu 1069/4, 958 01 Partizánske, Slovakia

**Keywords:** rubber, sulfur vulcanization, peroxide vulcanization, cross-linking, physical–mechanical characteristics, dynamical–mechanical analysis

## Abstract

The sulfur curing system, peroxide curing system and their combinations were applied for the cross-linking of unfilled and carbon black-filled rubber formulations based on ethylene-propylenediene-monomer rubber. The results demonstrated that the type of curing system influenced the course and shape of curing isotherms. This resulted in the change of curing kinetics of rubber compounds. The cross-link density of materials cured with combined vulcanization systems was lower than that for vulcanizates cured with the peroxide or sulfur system. Good correlation between the cross-link density as well as the structure of the formed cross-links and physical–mechanical characteristics of the cured materials was established. Both filled and unfilled vulcanizates cured with combined vulcanization systems exhibited a higher tensile strength and elongation at break when compared to their equivalents vulcanized in the presence of the peroxide or sulfur curing system. It can be stated that by proper combination of vulcanization systems, it is possible to modify the tensile behavior of vulcanizates in a targeted manner. On the other side, dynamical–mechanical properties were found not be significantly influenced by the curing system composition.

## 1. Introduction

Vulcanization or curing is a very important process in rubber technology. During this process, the plastics rubber compound changes into a highly deformable and elastic material called vulcanizate, by the formation of a three-dimensional cross-linked structure within the compound matrix. This structure is generated by chemical reactions between curing agents and functional groups of rubbers. A lot of vulcanization additives have been used for the cross-linking of rubber compounds such as sulfur curing systems, phenolic resins, organic peroxides, quinones, ureas, diamines, metal oxides, etc. The choice of vulcanization system is dependent mainly on the type of rubber or rubbers in the formulations and on the requirements of the final product properties. Vulcanization with sulfur-based systems and organic peroxides is still the most frequently used. 

Sulfur vulcanization is generally used for general purpose rubbers as well as specialty type elastomers containing double bonds. Sulfur is always used together with accelerators and activators. During sulfur vulcanization, sulfidic cross-links with different lengths (monosulfidic -S-, disulfdidic -S2-, and polysulfidic cross-links -Sx-, in which x = 3–6) are formed between polymer chain segments. Even though the sulfur vulcanization process has been known for more than 160 years, the mechanism and chemical pathways are very complex and are still a matter of ongoing research. Several mechanisms have been proposed as having ionic or radical character [1,2,3,4]. The main attributes of sulfur-based vulcanizates are good physical–mechanical properties, good tensile and tear strength, or good elastic and dynamic properties. The drawbacks are poor heat ageing stability and low resistance to thermo-oxidative ageing [5,6]. 

Organic peroxides are used to vulcanize both saturated and unsaturated rubbers. Peroxide vulcanization is a well-defined process proceeding via radical pathways. The first step is homolytic cleavage of organic peroxide at a high temperature with the formation of primary radicals. These fragments can split into secondary intermediates [7,8,9]. The formed radicals react with polymer chains to generate carbon–carbon bonds between them [10,11]. C–C cross-links have higher bonding energy when compared to sulfidic bridges, thus the main features of peroxide-cured materials are good resistance to elevated temperatures and ageing [2,12]. On the contrary, they exhibit worse tensile behavior and weaker dynamic characteristics. 

The cross-linking of rubber compounds with organic peroxides can be efficiently enhanced by the application of co-agents. Co-agents are organic substances with activated double bonds, which can react with rubber chains by addition and abstraction of hydrogen or by addition reactions only [13,14]. By grafting onto rubber chains, they can form extra cross-links which are dependent on the type of co-agent used [15]. Several benefits have been described for vulcanizates cured with peroxide and a co-agent compared to those treated only with organic peroxide, such as higher tensile and tear stress, higher hardness, higher modulus, enhanced dynamic properties or improved compression set, etc. [16,17,18,19].

Ethylene-propylene-diene-monomer rubber (EPDM) is a terpolymer comprised of ethylene and propylene structural units with randomly distributed non-conjugated diene such as 5-vinyl-2-norbornene, dicyclopentadiene or 5-ethylidene-2-norbornene. It has a saturated polymer backbone and pendant unsaturation in the diene monomer unit. EPDM belongs to the most versatile specialty type of rubbers. Its production has been continually increasing as it can be used for specialized as well as general applications. EPDM has very good electro-insulating properties, very good resistance to polar agents, water, and vapor. It has also very good resistance to oxygen, ozone, elevated temperature, and several chemicals. EPDM exhibits very good elastic properties at low temperatures and low compression set. However, its tensile strength is rather low. EPDM can also be effectively cured with the peroxide and sulfur vulcanization systems [8,20,21,22]. 

The peroxide vulcanization system, sulfur vulcanization system and combined vulcanization systems have been used for the cross-linking of virgin EPDM and EDPM filled with carbon black. As each curing system has both benefits and drawbacks, the main point was to combine the vulcanization systems by combining the patterns of the cross-links provided by each vulcanization system, thus suppressing the negatives and possibly highlighting the positives. Few studies have been devoted to the fabrication and investigation of properties of rubber compounds cured with combined sulfur and peroxide vulcanization systems. Nevertheless, the effect of co-agents in vulcanization systems has not been deeply considered. The results demonstrated that by combining both curing systems in suitable amount and proper composition, the characteristics of cured rubber compounds can be easily modified in the desired way. 

## 2. Experimental

### 2.1. Materials

Ethylene-propylene-diene-monomer rubber (EPDM, type KEP 570F) was provided by Kumho Polychem Co. Ltd., Yeosu, Republic of Korea. The rubber consisted of 70 wt.% ethylene units, 4.5 wt.% of 5-ethylidene-2-norbornene monomer units (ENB) and the rest were propylene structural units. Carbon black (CB, type Continex N330) was provided by Continental Carbon Co., Brussels, Belgium. The sulfur curing system consisted of activators—zinc oxide and stearic acid (Slovlak, Košeca, Slovakia), accelerator—N-cyclohexyl-2-benzothiazole sulfenamide (CBS, Duslo, Šaľa, Slovakia) and curing agent—sulfur (Siarkopol, Tarnobrzeg, Poland). The peroxide curing system consisted of dicumyl peroxide (DCP) in combination with trimethylolpropane trimethacrylate (TMPTMA) as a co-agent. They were supplied by Sigma-Aldrich, St. Louis, MO, USA. 

### 2.2. Methods

#### 2.2.1. Preparation and Curing of Rubber Compounds

Five types of rubber compounds based on virgin EPDM were fabricated and tested in this study. The first rubber compound was fabricated only by application of the peroxide vulcanization system, while in the last one, only the sulfur vulcanization system was introduced. In the remaining three types of rubber formulations, the mutual ratio of peroxide and sulfur vulcanization system was changed. The next five types of rubber formulations were fabricated with the same composition of curing systems, but carbon black in the constant amount 25 phr was used as a filler in each rubber compound. The carbon black rubber masterbatch was pre-compounded in a kneading machine, Buzuluk (Komárov, Czech Republic) in Vipo. a.s, Partizánske. The composition of unfilled and filled rubber compounds is summarized in Table 1 and Table 2. 

The rubber compounds were fabricated using a laboratory kneading machine, Brabender (Brabender GmbH & Co. KG, Duisburg, Germany) in a two-step mixing process. The temperature of the kneading chamber was 80 °C and the speed of the rotor was set to 55 rpm. First, rubber or rubber masterbatch was plasticated for 1 min. Zinc oxide and stearic acid were then added within the next 3 min and compounded. The mixed materials were subsequently cooled down in a calendar roll mill and put back in the kneading chamber. In the second step, the additives of sulfur curing (sulfur, accelerator) and/or the peroxide curing system (peroxide, co-agent) were added. The second step of mixing took 4 min at 55 rpm and 80 °C. In the final step, the compounds were cooled down and sheeted in the calendar. 

The vulcanization process of compounded formulations was performed at 180 °C and a pressure of 15 MPa by employing a hydraulic heated platens press, Fontijne (Fontijne Presses b.v., Vlaardingen, The Netherlands) based on a previously determined optimum cure time. 

#### 2.2.2. Determination of Curing Characteristics

The curing parameters of rubber formulations were evaluated from corresponding vulcanization curves, which were obtained by performing oscillatory measurements in moving die rheometer MDR 2000 (Alpha Technologies, Akron, OH, USA).

The evaluated characteristics were:

*M*—torque (dN.m)

∆*M* (dN.m)—the difference between the maximum and minimum torque (∆*M* = *M_H_* − *M_L_*).

*t_c_*_90_ (min)—optimum curing time 

*t_s_*_1_ (min)—scorch time

*R_v_* (min^−1^)—curing rate index, which is defined as:(1)$$R_{v}=\frac{100}{t_{c90}-t_{s1}}$$

*R_rev_* (min^−1^)—reversion rate index, which is defined as:(2)$$R_{rev}=\frac{100}{t_{rev}-t_{H}}$$

*t_rev_*—time at which *M_rev_* = *M_H_* − 0.02·Δ*M*

*t_H_*—time at which *M* = *M_H_*

#### 2.2.3. Determination of Cross-Link Density 

To calculate the cross-link density *ν*, small specimens of vulcanizates were fully immersed in xylene for 30 h at laboratory temperature. During this time, xylene diffuses into the samples and disrupts almost all physical bonds and interactions within the compound matrix, while chemical cross-links remain intact. The weight of samples was regularly detected every hour until the equilibrium swelling was reached. This means that no physical bonds were present in the rubber compound and free space was occupied by solvent. The concentration of chemical cross-links (i.e., the chemical cross-link density) is then possible to calculate. The Flory–Rehner equation for unfilled vulcanizates and the Flory–Rehner equation modified by Krause for carbon black-filled vulcanizates [23] was used to determine the cross-link density based on the equilibrium swelling state that was previously achieved.

#### 2.2.4. Investigation of Physical–Mechanical Characteristics

Zwick Roell/Z 2.5 (Zwick GmbH & Co. KG, Ulm, Germany) was employed to investigate the tensile behavior of vulcanizates. Dumbbell-shaped test specimens with a width of 6.4 mm, length of 80 mm, and thickness of 2 mm were used for measurements, while the cross-head speed of the appliance was set up to 500 mm·min^−1^. A durometer was used to detect the hardness of vulcanizates in Shore A.

#### 2.2.5. Determination of Dynamical–Mechanical Properties

The visco-elastic properties of vulcanizates were investigated by using a dynamic-mechanical thermal analyzer, DMTA MkIII (Mettler-Toledo GmbH, Greifensee, Switzerland). The samples were analyzed in tensile mode in a temperature range from −60 °C to 80 °C, at a frequency of 10 Hz, an amplitude of dynamic deformation 64 μm, and a static force 0.2 N. 

## 3. Results and Discussion

### 3.1. Curing Process 

The influence of curing systems on the vulcanization process of rubber compounds was evaluated based on the determination of vulcanization characteristics, which were calculated from vulcanization isotherms (Figure 1 and Figure 2). From Figure 1 it becomes apparent that the rubber compounds cured with the peroxide system (S0–1.5) exhibited the lowest scorch time *t_s_*_1_, and the torque stabilized at a constant value after reaching the maximum. The samples with a higher proportion of the peroxide system (S0.5–P1) and equivalent ratio of sulfur to peroxide (S0.75–P0.75) were found to have a low scorch time and were keeping “plato” after reaching maximum torque. On the other hand, rubber compounds with a higher ratio of the sulfur system (S1–P0.5 and S1.5–P0) demonstrated a longer scorch time, but after reaching the maximum torque, the reversion occurred. A similar behavior was possible to observe from curing isotherms of filled rubber compounds (Figure 2). From Figure 3 it becomes apparent that the lowest scorch time exhibited both unfilled and filled rubber compounds, cured only with the peroxide system (S0–P1.5). The higher the ratio of sulfur in combined vulcanization systems, the longer the scorch time. A similar effect was also demonstrated in work undertaken by Nikoleva et al. [24]. The highest *t_s_*_1_ exhibited the unfilled rubber compound cured only with the sulfur system (S1.5–P0). 

Peroxide cross-linking of rubber compounds is a relatively simple radical process, during which organic peroxides undergo homolytic cleavage by breaking the labile oxygen–oxygen bond at a vulcanization temperature. The peroxide fragments then immediately react with polymer functional groups to form cross-links between rubber chain segments [10,11,18,25]. Thus, the regulation of the scorch period during the peroxide vulcanization process can only be done by the type of peroxide and its dissociation rate at a curing temperature. On the other hand, sulfur vulcanization is a very intricate process running in several stages. The scorch time as well as the whole vulcanization process can be well-controlled by the composition of the sulfur vulcanization system, mainly by the amount and also the type of accelerator [2,26]. N-cyclohexyl-2-benzothiazole sulfenamide (CBS) belongs to the class called delayed action accelerators. During the curing process it prolongs the induction period, which is good for the processing of rubber compounds to the desired shapes without premature cross-linking. The main cross-linking phase is then very fast, which is more pronounced from both economical and time aspects [27]. This corresponds with the prolongation of the scorch time, with an increasing ratio of sulfur in both unfilled and filled rubber formulations. 

When comparing vulcanization isotherms of unfilled and filled rubber compounds, it becomes apparent that the curing process of rubber compounds with incorporated carbon black was faster, which indicates a steeper slope of curing isotherms in the main cross-linking phase. The calculated values of the optimum cure time *t_c_*_90_ (Figure 4) and curing rate index *R_v_* (Figure 5) clearly pointed out to the faster curing kinetics of filled rubber compounds. This can be attributed to carbon black that enhanced the heat transfer within the rubber compounds, which resulted in an acceleration of the curing process. The optimum cure time of unfilled rubber compounds increased with the increasing ratio of sulfur in combined vulcanization systems. The sample cured with the sulfur system (S1.5–P0) required more than 4 min longer for optimum cross-linking when compared to the rubber compound cured with the peroxide system (S0–P1.5). As also shown in Figure 4, the *t_c_*_90_ of filled rubber compounds was shorter than that of the unfilled formulations. There were recorded no significant changes of the optimum cure time in dependence on the curing system composition. The filled rubber compounds cured either with the peroxide (S0–P1.5) or sulfur (S1.5–P0) system were found to have very similar *t_c_*_90_. It can be stated that the influence of the curing system composition on scorch time and optimum cure time of the filled rubber systems was less visible in comparison with the unfilled equivalents. The highest curing rate index *R_v_* exhibited the filled rubber compound cured with the sulfur system (S1.5–P0), followed by the sample cured with the peroxide system (S0–P1.5). From Figure 5, it also becomes apparent that both filled and unfilled rubber compounds cured with combined vulcanization systems (S0.5–P1, S0.75–P0.75, S1–P0.5) demonstrated lower *R_v_* when compared to their equivalents cured with the peroxide or sulfur system. Similarly, both types of rubber compounds cured with the sulfur or peroxide curing system exhibited the highest difference between the maximum and minimum torque. The application of combined vulcanization systems caused the decrease in Δ*M*, although the differences in dependence on the curing system composition were not very significant. As also shown in Figure 6, higher Δ*M* values exhibited filled rubber compounds, which can again be attributed to the reinforcing filler. Carbon black stiffens and reinforces the rubber matrix and increases the viscosity of the compounds, which was reflected in the increase of minimum and maximum torque as well as in the increase of the torque difference. 

From curing isotherms, one can also see that some rubber compounds undergo reversion after reaching the maximum torque. Reversion is an undesired phenomenon, during which the destruction of the cross-links formed in the main vulcanization phase occurs. From Figure 7 it becomes obvious that the reversion rate index *R_rev_* showed an increasing trend with an increasing ratio of the sulfur curing system. The reason is ascribed to the structure of the formed cross-links. The application of the peroxide curing systems leads to the formation of carbon–carbon and multifunctional cross-links between rubber chains, which have high bonding energy and high thermal stability. On the other hand, sulfidic cross-links with different lengths (monosulfidic, disulfic, polysulfidic cross-links) are generated within the rubber compounds when they are cured in the presence of the sulfur curing systems. Sulfidic bonds have lower dissociation energy and are less thermally stable [28,29]. Therefore, thermal decomposition of sulfidic cross-links occurs with an increase in the cure time. This leads to a decrease in the cross-link density and viscosity of the cured rubber compounds, which is indicated as the decrease in torque in curing isotherms. As also shown in Figure 7, the higher reversion rate index exhibited rubber compounds with incorporated carbon black, which is more evident for samples with designations S1–P0.5 and S1.5–P0. Thus, it can be concluded that on one hand, carbon black leads to a faster curing process of rubber compounds. On the other hand, the CB-filled rubber compounds are more prone to reversion, probably due to increased heat transfer through the samples. 

### 3.2. Cross-Link Density

The dependences of cross-link density (Figure 8) on the curing system composition are in close correlation with the dependences of torque difference (Figure 6), clearly confirming a close relation between both characteristics. When considering unfilled vulcanizates, the highest cross-link density demonstrated the sample cured with the peroxide system (S0–P1.5), followed by the vulcanizate cured sulfur system (S1.5–P0). The cross-link density of vulcanizates cured with combined vulcanization systems (S0.5–P1, S0.75–P0.75, S1–P0.5) was found to increase with the increasing ratio of sulfur to peroxide. The similar results were also recorded for filled vulcanizates, but the highest cross-link density was determined in the structure of the sample cured with the sulfur system. The unfilled vulcanizate cured with the peroxide system exhibited a higher cross-link density when compared to its filled equivalent. When applying the sulfur curing system and combined peroxide and sulfur curing systems, the higher cross-linking degree exhibited vulcanizates with incorporated carbon black. It also becomes apparent that differences between cross-link densities of unfilled and filled vulcanizates became higher with the increasing amount of sulfur in vulcanization systems. In general, carbon black-filled rubber systems exhibit a higher cross-link density when compared to their unfilled equivalents. This is because rubber chains in the proximity of CB particles are strongly physically adsorbed or chemisorbed on the filler, and they behave like a polymer in a glassy state. The strongly bonded rubber-filler layers are insoluble in the solvents used for the determination of cross-link density, and thus contribute to the apparent increase of the cross-linking degree. As shown in Figure 8, with the increase of the peroxide to sulfur ratio, the differences in cross-link density between both vulcanizate types became lower, while the unfilled vulcanizate with designation S0–P1.5 was found to have a higher cross-linking degree when compared to the filled counterpart. The reason might be attributed to the fact that acidic substances, like some types of carbon black, can cause heterolytic decomposition of the peroxide with formation of ions and not radicals, which are not effective in the cross-link formation. The results might suggest that unfilled EPDM-based vulcanizates are more convenient peroxide curing systems, while CB-filled counterparts seems to be more suitable for sulfur curing systems. However, it must be remarked that the values of cross-link density are not a determining step for the selection of the vulcanization system. Instead, the structure of the formed cross-links related to the properties of vulcanizates and their thermo-oxidative stability should be under consideration. 

As already outlined, during peroxide vulcanization, dicumyl peroxide first undergoes homolytic heat-induced cleavage, with the formation of primary cumyloxy radicals that are dissociated into secondary methyl radicals [30]. Acetophenone is released as a byproduct from thermal decomposition of the peroxide. Both cumyloxy and methyl radicals are supposed to be efficient during peroxide cross-linking of rubbers [18,31]. EPDM has saturated the main polymer chain with randomly distributed non-conjugated diene monomer having pendant unsaturation. Peroxide cross-linking of EPDM is relatively efficient. Radical fragments from peroxide can abstract hydrogen atoms from the main elastomer backbone yielding alkyl macroradicals, but also from diene units, in which the H-atom is situated in an allylic position (allyl macromolecular radicals). Cross-linking occurs through two pathways. The recombination of alkyl and allyl macroradicals leads to the formation of allyl/allyl, alkyl/alkyl and alkyl/allyl cross-link combinations. EPDM alkyl and allyl macroradicals can also add to the double bonds in diene monomer units. For steric reasons, the macromolecular radical thus formed would not propagate with another diene unit, but probably terminates via hydrogen transfer with the formation of allyl/alkene and alkyl/alkene cross-links [32,33,34,35,36,37]. 

Dicumyl peroxide was used in combination with trimethylolpropane trimethacrylate. During co-agent-assisted peroxide vulcanization, not only are carbon–carbon bonds formed within the rubber matrix, but multifunctional cross-links are also generated through grafting of co-agents between rubber chains segments [2,19,35,38]. In general, co-agents are low molecular weight organic molecules that have a high reactivity with free radicals. They boost the cross-linking efficiency of the curing process with organic peroxides and tend to increase the cross-link density. Most of them have been reported to homopolymerize and graft to macroradicals, and thus form co-agent bridges between the polymer chain segments as extra cross-links [13,14,39,40,41]. This subsequently leads to the property improvements of the final products. 

Chemism and mechanism of sulfur vulcanization is very intricate and is still not comprehensibly clarified. It involves a series and complex parallel of consecutive reactions which may have additive, substitutive or even elimination characters. Not only can the input materials take part in the process, but so can products of their transformations and intermediates. The sulfur cross-linking mechanism for EPDM has been reported to be similar, which is generally accepted for diene rubbers [2,42]. In the first stage, the accelerator together with activators form the transition complex, which reacts with sulfur to form an active sulfurating agent. The complex thus formed substitutes the hydrogen atom in the allylic position of ENB and is attached to the rubber via a sulfur bridge, forming cross-link precursors [43]. In the second stage, a primary cross-linked network is formed with dominance of polysulfidic bonds. The last stage relates to the restructuralization of the primary network by lowering the number of sulfur atoms in sulfidic bridges and modification of rubber chains, after which the final cross-linked vulcanizate network is generated. 

As seen in Figure 8, the cross-link density of vulcanizates cured with combined vulcanization systems was lower when compared to the samples cured either with the peroxide or sulfur-curing system. For both unfilled and filled vulcanizates, the lowest cross-link degree demonstrated the sample with designation S0.5–P1. It is likely that interactions between the additives of both curing systems occurred which resulted in the decrease of the effective concentration of peroxide and sulfur fragments, which were then not efficient during the curing process. 

### 3.3. Physical–Mechanical Properties 

The dependences modulus M100 (Figure 9) on curing systems correlates with the dependences of the cross-link density (Figure 8). The highest modulus M100 exhibited the filled vulcanizate cured with the sulfur system (S1.5–P0) with the highest cross-link density, followed by the vulcanizate cured in the presence of the peroxide system (S0–P1.5). The modulus of the filled vulcanizates cured with the combined vulcanization systems (S0.5–P1, S0.75–P0.75, S1–P0.5) showed an increasing tendency, with an increasing sulfur to peroxide ratio. The influence of the curing system composition on the modulus of unfilled vulcanizates was much less visible, as the values of M100 were very similar for each tested cured rubber compound. The higher modulus of filled vulcanizates can be ascribed to their higher cross-linking degree (with the exception of the sample S0–P1.5) and to the reinforcing effect of carbon black. The highest cross-link density of the filled vulcanizate cured with the sulfur system (S1.5–P0) was also reflected in the highest hardness of the corresponding vulcanizate (Figure 10). The vulcanizates cured with the combined vulcanization systems were found to have a lower hardness when compared to the equivalent rubber compounds cured only with the sulfur or peroxide system. Again, the influence of the curing system composition on the hardness of unfilled vulcanizates was less pronounced. 

One can also observe a certain correlation between the dependences of the cross-link density and elongation at break of vulcanizates. From Figure 11, it becomes obvious that the highest elongation at break demonstrated both unfilled and filled vulcanizates, with designation S0.5–P1 having the lowest cross-linking degree. The higher the cross-link density, the greater the restriction of the elasticity and mobility of the rubber chain segments, which had a negative impact on elongation at break. Thus, it becomes apparent that the lowest elongation at break reached the vulcanizates cured with the sulfur (S1.5–P0) or peroxide (S0–P1.5) system with the highest cross-link density. The higher elongation at break exhibited vulcanizates with incorporated carbon black despite their higher cross-link density. 

When looking at the graphical dependence of the curing system composition on tensile strength (Figure 12), one can see that the highest values exhibited unfilled and filled vulcanizates, which cured in the presence of 1 phr DCP and 0.5 phr sulfur (S0.5–P1). On the other hand, the lowest tensile properties demonstrated the vulcanizates cured only with the sulfur or peroxide system. It would seem that the tensile behavior of vulcanizates is also proportional to the cross-link density. However, the tensile strength is a complex property which is dependent not only on the cross-linking degree, but mainly on the type and structure of the formed cross-links, and also on the type and amount of the filler [44]. It becomes apparent that CB-filled vulcanizates cured with combined vulcanization systems (S0.5–P1, S0.75–P0.75, S1–P0.5) exhibited much higher tensile strength in comparison with their counterparts, cured only in the presence of the peroxide (S0–P1.5) or sulfur (S1.5–P0) system. The tensile strength of the vulcanizate with designation S0.5–P1 (over 32 MPa) was more than two times higher in comparison with the vulcanizates S0–P1.5 (over 15 MPa) and S1.5–P0 (13.5 MPa). The unfilled vulcanizates cured with the combined vulcanization systems also demonstrated a higher tensile strength when compared to vulcanizates with designations S0–P1.5 and S1.5–P0. However, the differences in their values were not so pronounced as in the case of the filled vulcanizates. The vulcanizates cured with the sulfur (S1.5–P0) or peroxide (S0–P1.5) system reached almost the same tensile strength (2.2 MPa). When the peroxide system was combined with the sulfur system in the ratio S0.5–P1, the tensile strength increased up to 4 MPa. It also becomes apparent that a much higher tensile strength demonstrated filled vulcanizates due to the reinforcing of carbon black. 

The obtained results pointed out to some synergic effect of both curing systems. By mutual combination of both systems, the pattern of sulfidic cross-links, carbon–carbon cross-links, and multifunctional cross-links from the co-agent are combined, which results in the modification of physical–mechanical characteristics of the final products. Based upon the achieved experimental outputs, it can also be stated that the influence of vulcanization systems is more evident for filled vulcanizates. Thus, the type of the filler must also be considered when applying vulcanization systems for the cross-linking of rubber compounds. 

### 3.4. Dynamical–Mechanical Analysis

Dynamical–mechanical analysis was carried out to evaluate the influence of the composition of curing systems on the visco-elastic properties of vulcanizates. Samples cured only with sulfur, peroxide, and the combined sulfur and peroxide systems in their equivalent ratio were chosen for the measurement. 

The results revealed that storage *G*′ as well as loss *G*″ modulus were not significantly influenced by the composition of the curing systems. Thus, no meaningful change in loss factor *tan δ* was recorded in dependence on the curing system composition. The temperature dependences of *tan δ* for unfilled and filled vulcanizates are graphically illustrated in Figure 13 and Figure 14, while the calculated values at specific temperatures are summarized in Table 3 and Table 4. As shown, almost no change of loss factor was recorded in dependence on the curing system composition for both types of vulcanizates. The higher the temperature, the lower *tan δ* was. At very low temperatures, the higher loss factor exhibited unfilled vulcanizates. With an increase in temperature, the differences in *tan δ* for both vulcanizate types became smaller. The peak maximum in the loss factor temperature dependences corresponds to the glass transition temperature *T_g_*. In general, the *T_g_* is dependent mainly on the type of rubber or rubbers in the formulations, but can also be influenced by other factors such as the presence of additives, mostly plasticizers or fillers, cross-link density, etc. The type and structure of the formed cross-links might also have an influence on the glass transition temperature [45]. From Figure 13 and Table 3, it becomes apparent that *T_g_* of the unfilled rubber systems was almost the same regardless of the composition of the curing system. The same statement can also be applied for filled vulcanizates. As shown in Figure 14 and Table 4, the glass transition temperature for filled vulcanizates moved only in the low range of the experimental values. It can also be stated that the presence of carbon black in rubber formulations did not influence the *T_g_*. Although the *T_g_* of filled vulcanizates was about 1–1.5 °C lower when compared to unfilled equivalents, the decline can be considered negligible. 

## 4. Conclusions

The results revealed the higher the amount of sulfur in mixed curing systems, the longer the scorch time. The optimum cure time of unfilled rubber compounds showed an increasing tendency with an increasing ratio of sulfur, while the optimum cure time of filled rubber formulations seems not to be so dependent on the curing system composition. Rubber formulations with higher amounts of the sulfur system were more prone to reversion after reaching maximum torque. Faster vulcanization was observed for filled rubber compounds due to carbon black, which enhanced the heat transfer through the cured samples. The highest cross-link density exhibited both unfilled and filled vulcanizates, cured only with the sulfur or peroxide system. The cross-link density of vulcanizates cured with the combined sulfur/peroxide systems showed an increasing trend, with an increasing sulfur to peroxide ratio. The dependences of hardness, modulus and elongation at break were in close correlation with the dependences of the cross-link density. The tensile strength of vulcanizates cured in the presence of the combined vulcanization systems was higher when compared to their equivalents cured only with the sulfur or peroxide system. This points out the fact that by mutual combinations of curing systems, the pattern of sulfidic cross-links, carbon–carbon bonds, and multifunctional cross-links from co-agents are suitably combined, which resulted in an improvement in the tensile behavior of vulcanizates. On the other hand, no considerable effect of curing system compositions on dynamic–mechanical characteristics was recorded.

## Figures and Tables

**Figure 1 materials-16-05596-f001:**
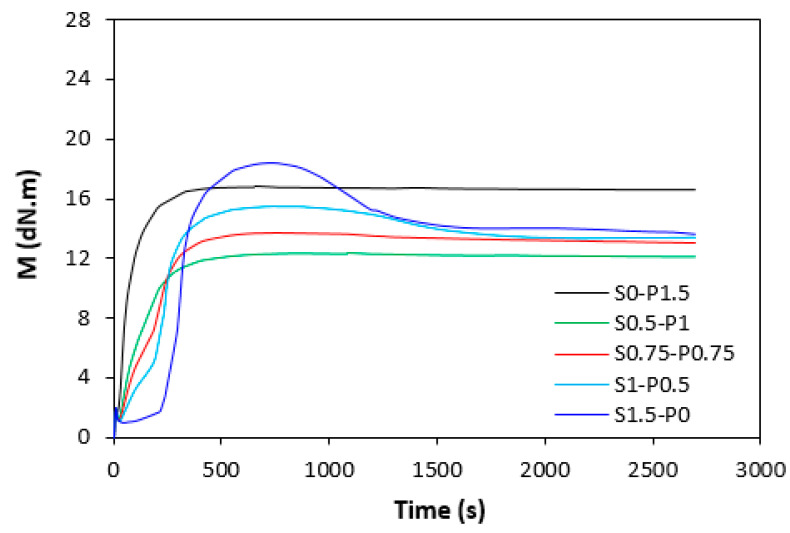
Dependence of torque *M* on time during vulcanization process at 180 °C for unfilled rubber compounds.

**Figure 2 materials-16-05596-f002:**
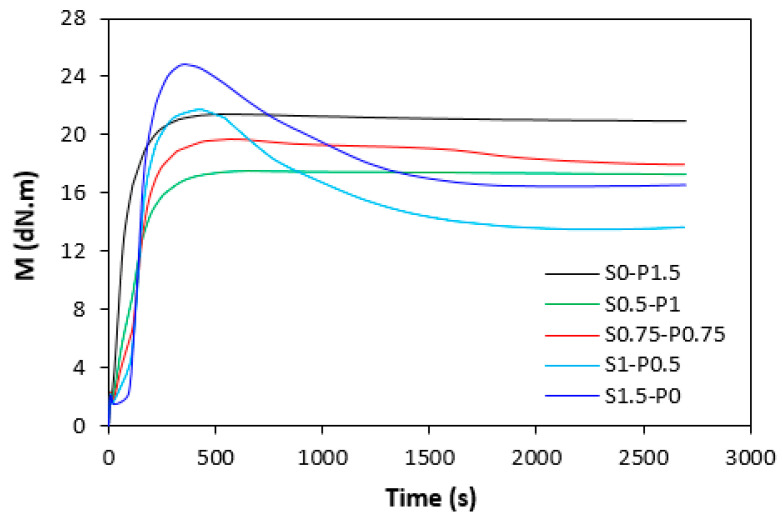
Dependence of torque *M* on time during vulcanization process at 180 °C for filled rubber compounds.

**Figure 3 materials-16-05596-f003:**
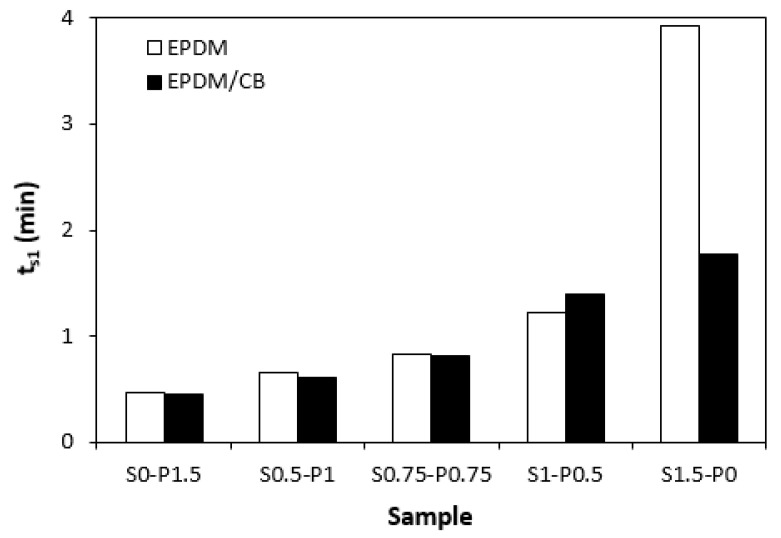
Influence of composition of vulcanization system on scorch time *t_s_*_1_ of rubber compounds.

**Figure 4 materials-16-05596-f004:**
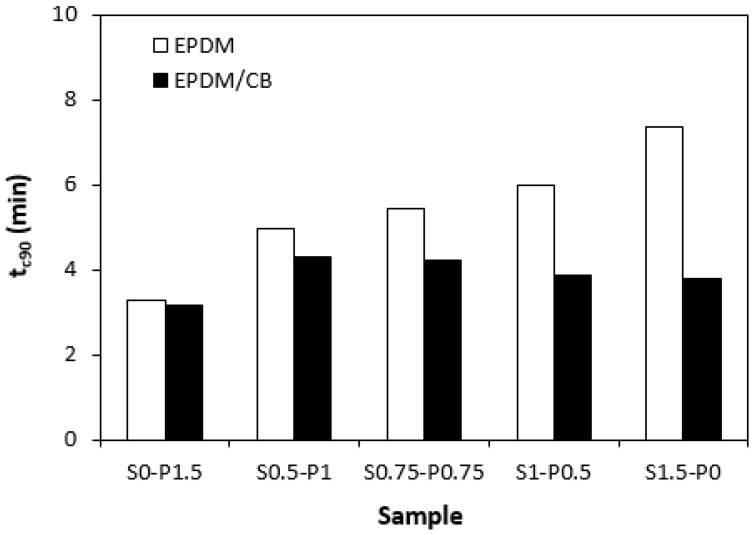
Influence of composition of vulcanization system on optimum cure time *t_c_*_90_ of rubber compounds.

**Figure 5 materials-16-05596-f005:**
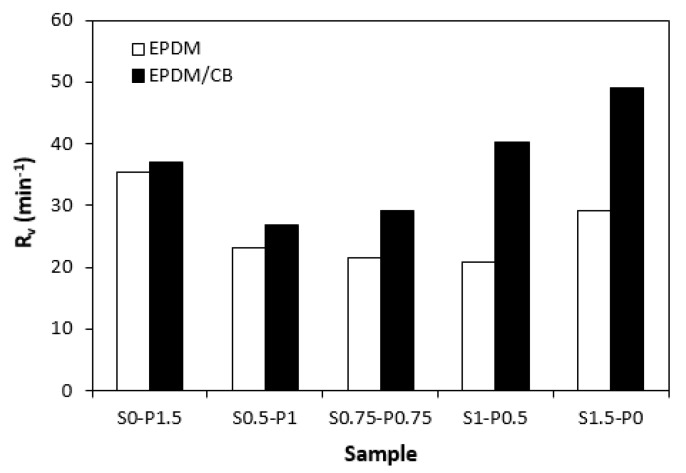
Influence of composition of vulcanization system on curing rate index *R_v_* of rubber compounds.

**Figure 6 materials-16-05596-f006:**
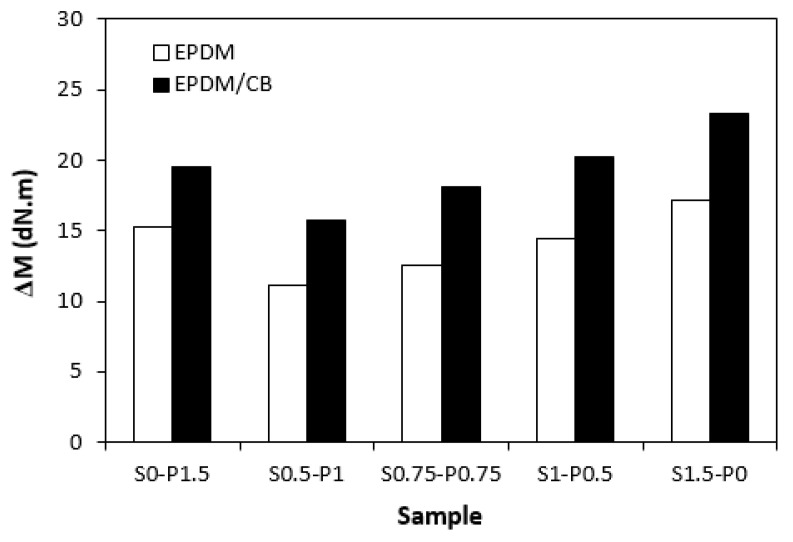
Influence of composition of vulcanization system on torque difference Δ*M* of rubber compounds.

**Figure 7 materials-16-05596-f007:**
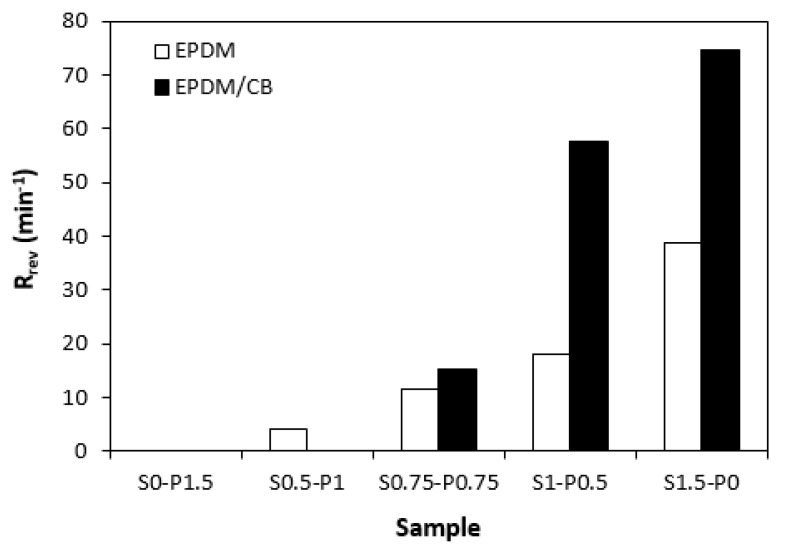
Influence of composition of vulcanization system on reversion rate index *R_rev_* of rubber compounds.

**Figure 8 materials-16-05596-f008:**
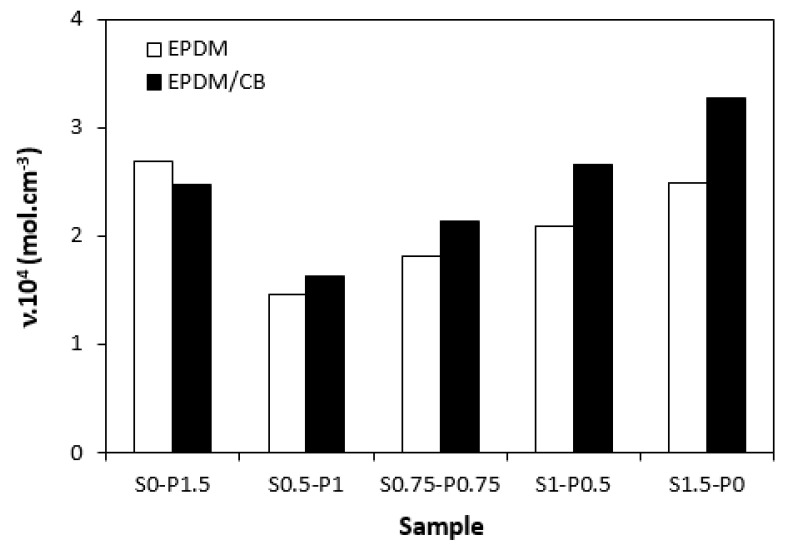
Influence of composition of vulcanization system on cross-link density *ν* of vulcanizates.

**Figure 9 materials-16-05596-f009:**
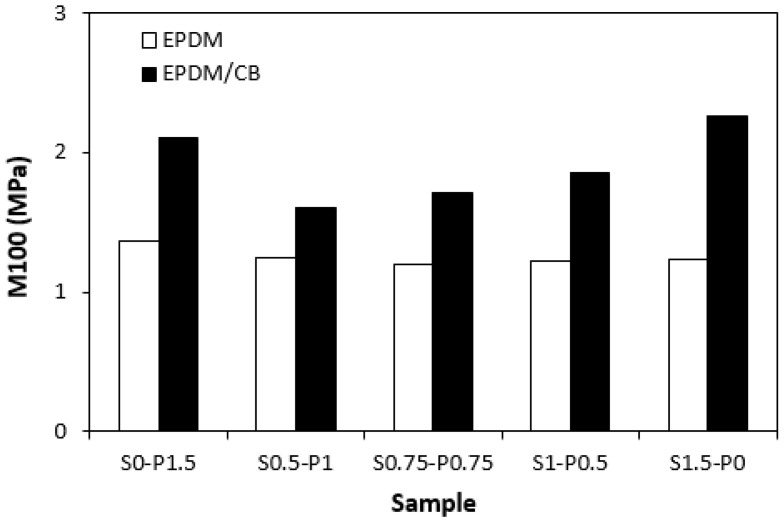
Influence of composition of vulcanization system on modulus M100 of vulcanizates.

**Figure 10 materials-16-05596-f010:**
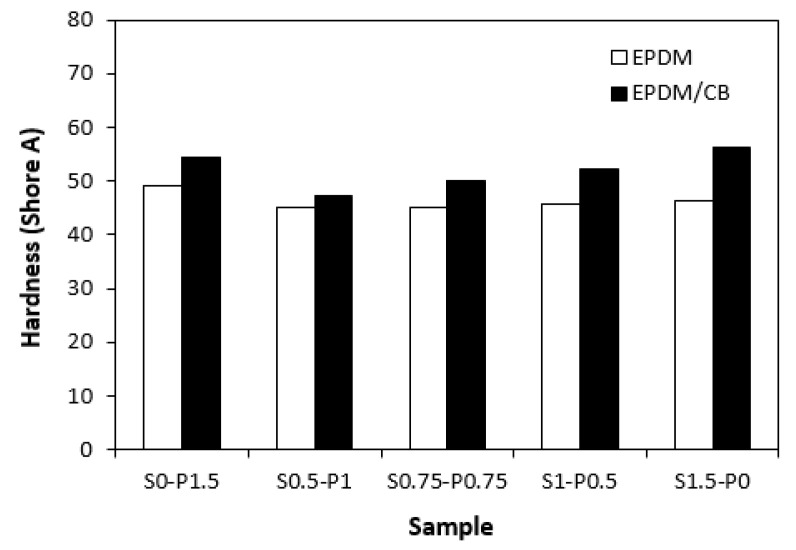
Influence of composition of vulcanization system on hardness of vulcanizates.

**Figure 11 materials-16-05596-f011:**
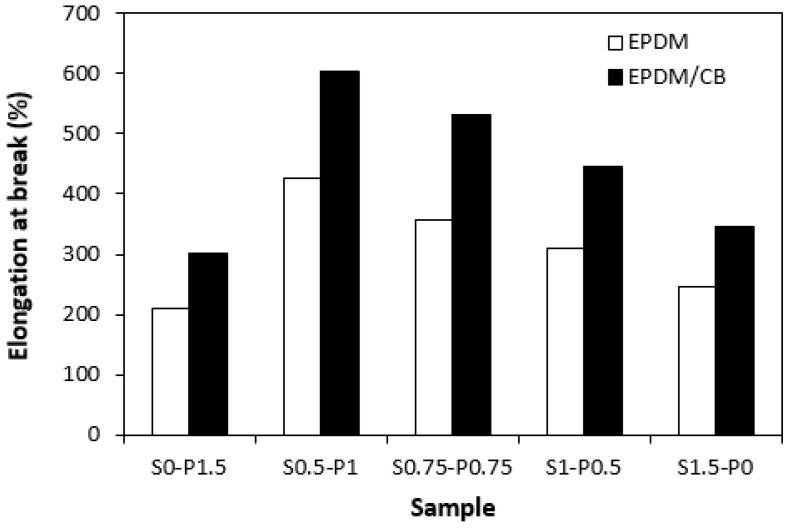
Influence of composition of vulcanization system on elongation at break of vulcanizates.

**Figure 12 materials-16-05596-f012:**
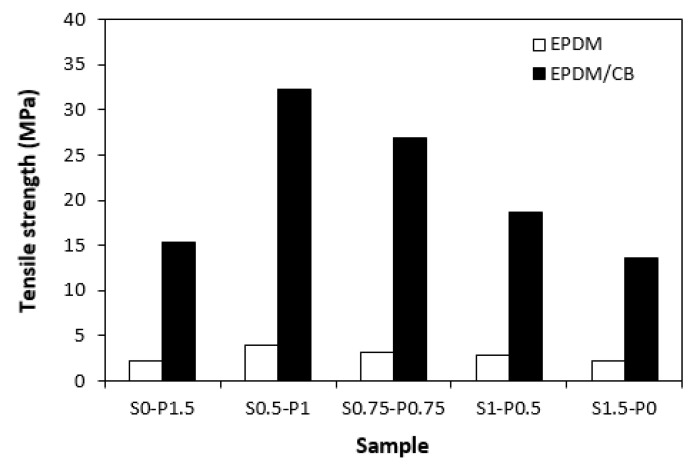
Influence of composition of vulcanization system on tensile strength of vulcanizates.

**Figure 13 materials-16-05596-f013:**
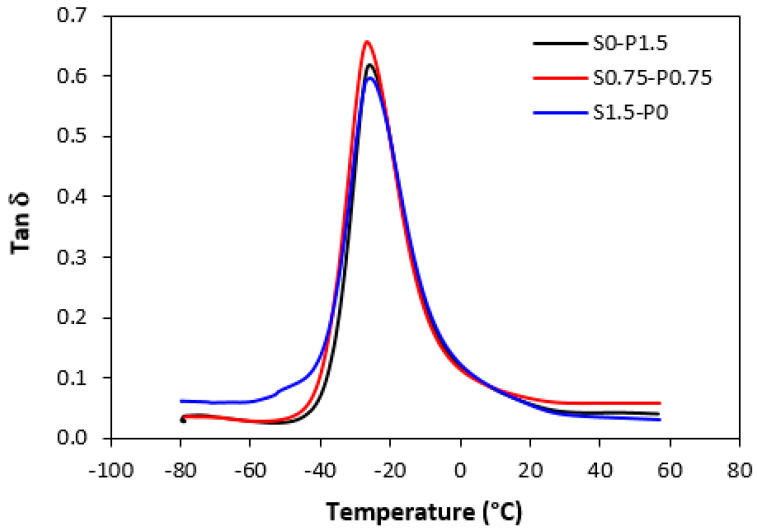
Temperature dependences of loss factor *tan δ* for unfilled vulcanizates.

**Figure 14 materials-16-05596-f014:**
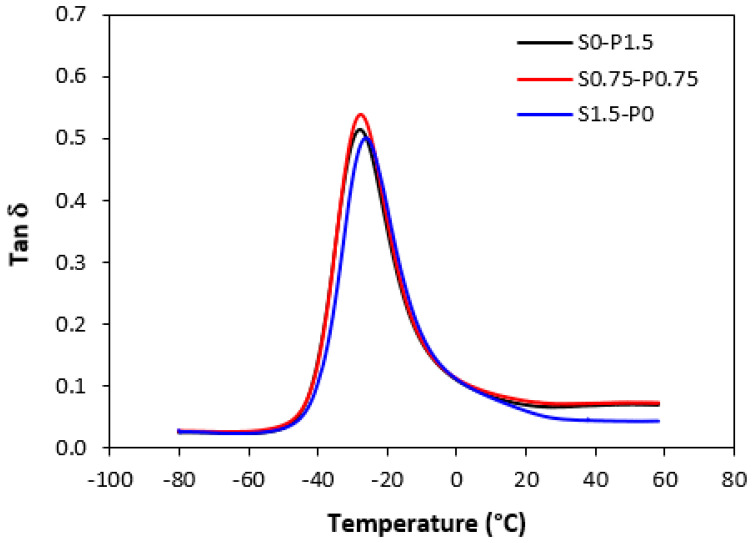
Temperature dependences of loss factor *tan δ* for filled vulcanizates.

**Table 1 materials-16-05596-t001:** Composition of unfilled rubber compounds in phr and their designation.

	S0–P1.5	S0.5–P1	S0.75–P0.75	S1–P0.5	S1.5–P0
EPDM	100	100	100	100	100
ZnO	0	1	2	3	4
Stearic acid	0	0.5	1	1.5	2
Sulfur	0	0.5	0.75	1	1.5
CBS	0	1	2	3	4
DCP	1.5	1	0.75	0.5	0
TMPTMA	4	3	2	1	0

**Table 2 materials-16-05596-t002:** Composition of filled rubber compounds in phr and their designation.

	S0–P1.5	S0.5–P1	S0.75–P0.75	S1–P0.5	S1.5–P0
EPDM	100	100	100	100	100
CB	25	25	25	25	25
ZnO	0	1	2	3	4
Stearic acid	0	0.5	1	1.5	2
Sulfur	0	0.5	0.75	1	1.5
CBS	0	1	2	3	4
DCP	1.5	1	0.75	0.5	0
TMPTMA	4	3	2	1	0

**Table 3 materials-16-05596-t003:** Glass transition temperature *T_g_* and loss factor *tan δ* for unfilled vulcanizates.

Sample	*Tg* (°C)	*tan δ* (−20 °C)	*tan δ* (0 °C)	*tan δ* (20 °C)	*tan δ* (50 °C)
S0–P1.5	−26	0.50	0.12	0.06	0.04
S0.75–P0.75	−26.6	0.50	0.12	0.07	0.06
S1.5–P0	−25.8	0.50	0.12	0.06	0.03

**Table 4 materials-16-05596-t004:** Glass transition temperature *T_g_* and loss factor *tan δ* for filled vulcanizates.

Sample	*Tg* (°C)	*tan δ* (−20 °C)	*tan δ* (0 °C)	*tan δ* (20 °C)	*tan δ* (50 °C)
S0–P1.5	−27.9	0.35	0.11	0.07	0.07
S0.75–P0.75	−27.6	0.38	0.11	0.08	0.07
S1.5–P0	−26.3	0.39	0.11	0.06	0.04

## Data Availability

Data Availability Statements are available in section “MDPI Research Data Policies” at https://www.mdpi.com/ethics (accessed on 29 July 2023).

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
