# Peer review of "Combined Sulfur and Peroxide Vulcanization of Filled and Unfilled EPDM-Based Rubber Compounds"

_materials, 2023, doi:10.3390/ma16165596_

Round 1

Reviewer 1 Report

I wish to congratulate authors for their work; very nice having the opportunity to review this manuscript. The authors have prepared the manuscript very well, the methods are clear, and the discussion has been comprehensive. Conclusion is based on the findings.

For some minor revisions, please find below my comments:

1.     Abstract should contains more quantitative data.

2.     Introduction is too much explaining the theories. I suggests author should emphasize the state-of-the-art and explicitly state the novelty of their study.

3.     Quantitative data such as curing rate, tensile, elongation, and so on.. should be analyzed statistically and the mean of triplicated data should be presented.

4.     Authors may opt to combine the figures. Twelve figures are a lot.

Please check again for typing errors.

Author Response

-

Reviewer 2 Report

The manuscript describes the vulcanization of EPDM rubber. The authors studied the effect of the vulcanizing agent, sulphur or peroxide, and the filler carbon black on the curing process  and thermomechanical properties of vulcanizates. The paper is well arranged, the experimental results are properly discussed and the corresponding conclusions are well supported.

There is however one point to be explained.

l.359      The authors observe “certain correlation between the dependences of cross-link density and elongation at break of vulcanizates…. The higher was the crosslink density, the more was restricted the elasticity and mobility of rubber chain segments, which had negative impact on elongation at break…

 The lowest elongation at break reached the vulcanizates cured with sulfur (S1.5-P0) or peroxide (S0- P1.5) system with the highest cross-link density.”

However, the vulcanizates with incorporated carbon black show higher elongation at break despite their higher cross-link density and higher modulus (cf. Figs. 7,8,11). Why ?

The difference in tensile strength of filled and unfilled vulcanizates (Fig.12) is very impressive in contrast to relatively small differences of other mechanical properties (modulus, hardness, elongation at break).

Yes, the tensile strength is the complex property. I suggest to include the figure of stress-strain curves that could help to better understand this difference. Such a figure should explain the opposite trends in dependence of tensile strength and modulus, crosslinking density and hardness on composition.

Moreover, caption in Fig.12 claims by mistake the influence of glycerol content on tensile strength of vulcanizates.

The paper is of a high scientific quality and brings a new knowledge in the topic. I recommend the minor revision before publication.

Author Response

-

Reviewer 3 Report

This is an interesting paper on the effect of cocuring of EPDM with sulfur and dicumyl peroxide. It is recommended to accept this paper for publication after some revision on the basis of comments below.

COMMENTS

1.

The authors are suggested to study the existing literature in more details, and at least the following recent publications should be mentioned and cited in their manuscript:

Rahman Parathodika, A.; Raju, A. T.; Das, M.; Bhattacharya, A. B.; Neethirajan, J.; Naskar, K. Exploring hybrid vulcanization system in highmolecular weight EPDM rubber composites: A statistical approach. J. Appl. Polym. Sci. 2022, 139, e52721.

Kruzelák, J.; Hlozeková, K.; Kvasnicáková, A.; Tomanová, K.; Hudec, I. Application of Sulfur and Peroxide Curing Systems for Cross-Linking of Rubber Composites Filled with Calcium Lignosulfonate. Polymers 202214, 1921.

Parathodika, A. R.; Sreethu, T. K.; Maji, P.; Susoff, M.; Naskar, K. Influence of molecular and crosslink network structure on vulcanizate properties of EPDM elastomers. Express Polym. Lett. 2023, 17, 722-737.

Nikolova, S.; Mihaylov, M.; Dishovsky, N. Mixed peroxide/sulfur vulcanization of ethylene-propylene terpolymer based composites. curing characteristics, curing kinetics and mechanical properties. J. Chem. Technol. Metal. 2022, 57, 881-894.

2.

Did the authors carried out control experiments with EPDM in the absence of any vulcanization additive? Such data should be obtained and reported in this study. It is well-known that EPDM can undergo crosslinking without any curing additive by the process applied in this study, that is, at 180 C.

3.

In line 119, the authors write that the heating time at 180 C is “based on previously determined optimum cure time”. However, neither reference nor curing time data are provided. In order to have reproducible experiments by any scientists and engineers, the authors should provide exact experimental conditions with all the experimental details, including curing time in this case for all the applied formulations. This is the most basic requirement of a scientific, technological publication.

4.

It is unclear and wrong using a period (dot) instead of a dot as a multiplier in the middle between the multiplied parameters. This should be corrected in the full text and the axes of figures as well.

5.

The meaning of “M” is not given. This should be provided.

6.

In Figures 1 and 2, at least the vulcanization temperature should be provided in the figure caption.

7.

For the dynamical-mechanical analysis, the results as control for the pure EPDM, including the Tg as well, should be also provided.  

8.

The authors should give an explanation on the nearly constant Tg of the crosslinked EPDM independent of the used vulcanizing agents, their concentrations and crosslinking density. This is an unexpected result.

9.

The format of references is not composed according to the requirement of the journal. This should be corrected by the authors.

Author Response

-
